# Highly Efficient Solution-Processed Blue Phosphorescent Organic Light-Emitting Diodes Based on Co-Dopant and Co-Host System

**DOI:** 10.3390/molecules27206882

**Published:** 2022-10-14

**Authors:** Jingyu Wang, Jiaxuan Yin, Xiaofang Li, Zhe Wang, Ruixia Wu, Liang Zhou

**Affiliations:** 1State Key Laboratory of Rare Earth Resource Utilization, Changchun Institute of Applied Chemistry, Chinese Academy of Sciences, Changchun 130022, China; 2School of Applied Chemistry and Engineering, University of Science and Technology of China, Hefei 230027, China

**Keywords:** phosphorescent organic light-emitting diodes, co-dopant structure, solution process, co-host system, microcavity effect

## Abstract

The low-lying HOMO level of the blue emitter and the interfacial miscibility of organic materials result in inferior hole injection, and long exciton lifetime leads to triplet-triplet annihilation (TTA) and triplet-polaron annihilation (TPA), so the efficiencies of blue phosphorescent organic light-emitting diodes (PhOLEDs) are still unsatisfactory. Herein, we design co-host and co-dopant structures to improve the efficiency of blue PhOLEDs by means of solution processing. TcTa acts as hole transport ladder due to its high-lying HOMO level, and bipolar mCPPO1 helps to balance carriers’ distribution and weaken TPA. Besides the efficient FIr6, which acts as the dominant blue dopant, FCNIrPic was introduced as the second dopant, whose higher HOMO level accelerates hole injection and high triplet energy facilitates energy transfer. An interesting phenomenon caused by microcavity effect between anode and cathode was observed. With increasing thickness of ETL, peak position of electroluminescence (EL) spectrum red shifts gradually. Once the thickness of ETL exceeded 140 nm, emission peak blue-shifts went back to its original position. Finally, the maximum current efficiency (CE), power efficiency (PE), and external quantum efficiency (EQE) of blue phosphorescent organic light-emitting diode (PhOLED) went up to 20.47 cd/A, 11.96 lm/W, and 11.62%, respectively.

## 1. Introduction

As a new solid-state light source, organic light-emitting diodes (OLEDs) are expected to be used in flat-panel displays and lighting applications, so they have been widely discussed by researchers [1,2,3,4,5,6,7]. Vacuum thermal evaporation and solution process are the two main technologies for the fabrication of OLEDs [8]. Compared with the vacuum evaporation process, the solution process is easier in terms of adjusting the proportion of host and guest materials in the doping system and is superior in large-scale production [9,10,11,12,13]. However, it is difficult to achieve efficient multi-stacked structures due to miscibility between organic layer interfaces in solution-processed devices and the wide band gaps (~3 eV) of phosphorescent materials which result in inferior hole injection. As such, triplet excitons are easily quenched by redundant carriers (TPA process) [14,15,16]. In addition, the long triplet excitons lifetimes (~μs) of phosphorescent materials lead to excitons accumulation, which would greatly increase the quenching probability of triplet excitons (TTA process) [17,18,19], so the solution-processed blue phosphorescent devices display relatively lower efficiency and inferior operation stability. Therefore, the lack of high-performance blue devices seriously restricts the market application of OLEDs.

At present, various methods have been proposed to improve device performances. One efficient strategy is designing a co-host system consisting of hole-transporting material and either electron-transporting or bipolar material to enhance carriers’ balance and broaden recombination zone (RZ), thereby reducing driving voltage and improving device performances [20,21,22,23,24]. For example, Deepak and his colleagues prepared a co-host device by blending hole-transporting-type host material 4,4′,4′′-Tris(N-3-methylphenyl-N-phenyl-amino)-triphenylamine (m-MTDATA) and bipolar host material DT-320, which exhibited PE of 72.6 and 62.9 lm/W at 100 and 1000 cd/m^2^, respectively [25]. In addition, another feasible strategy is introducing sensitizers or dopants to realize efficient energy transfer [26,27,28,29,30,31,32]. Baldo and co-workers used green phosphor fac-Tris(2-phenylpyridine)iridium(III) (Ir(ppy)_3_) as a sensitizer to facilitate the transfer of triplet energy to red fluorescent emitter 4-(Dicyanomethylene)-2-methyl-6-julolidyl-9-enyl-4H-pyran (DCM2) and realized significantly higher DCM2 emission efficiency [33]. Seung et al. selected double dopants of FIrpic and FIr6 to obtain the device with maximum CE of 24.8 cd/A and slow efficiency roll-off, and they proved the efficacy of the second dopant in stabilizing carriers’ balance within the light-emitting layer (EML) and enhancing the energy transfer between two dopants [34].

In this work, we combined the mentioned two strategies by adopting a co-host system composed of bipolar material 9-(3-(9H-Carbazol-9-yl)phenyl)-3-(diphenylphosphor-yl)-9H-carbazole (mCPPO1) and hole-transporting material 4,4′,4′′-Tris(carbazol-9-yl)triphenylamine (TcTa) and introducing the second dopant Bis(3,5-difluoro-4-cyano-2-(2-pyridyl)phenyl-(2-carboxypyridyl)iridium(III) (FCNIrPic) to facilitate carriers’ balance as well as reduce exciton quenching, thus improving the efficiency of PhOLED. Subsequently, the roles of co-host and co-dopant in improving device performances were discussed in detail. Moreover, an interesting phenomenon, whereby the thickness of the electron transport layer (ETL) could significantly influence both efficiency and EL spectrum, was observed and interpreted. Finally, the maximum CE, PE, and EQE of optimized blue PhOLED device went up to 20.47 cd/A, 11.96 lm/W, and 11.62%, respectively.

## 2. Results and Discussion

The energy levels diagram of designed devices and the molecular structures of used materials are shown in Figure 1. In this case, PEDOT: PSS film was spin-coated on an ITO-coated glass substrate as the hole injection layer (HIL). Meanwhile, 1,3,5-Tri[(3-pyridyl)-phen-3-yl]benzene (TmPyPB), which exhibits high electron mobility and low-lying highest occupied molecular orbital (HOMO) level, was used as the electron-transporting and hole-blocking material. Bis(2,4-difluorophenylpyridinato)tetrakis(1-pyrazolyl)borate iridium(III) (FIr6) is a deep-blue phosphorescent material with 73% photoluminescence quantum efficiency (PLQY) and high thermal stability, so we selected it as the emitter [35]. To construct high-performance PhOLED, the key is to select host and phosphorescent materials with matched energy levels and triplet energies [36]. Because of the high triplet energy (2.72 eV) of FIr6, it is difficult to select appropriated host materials with high enough triplet energy to prevent the reverse energy transfer from dopant to host molecules [37]. In addition, the low-lying HOMO energy level of FIr6 results in inferior hole injection, thus causing unbalanced carriers’ distribution within EML. Therefore, we adopt a co-host system based on bipolar material mCPPO1 and hole-transporting material TcTa. The triplet energies of mCPPO1 (3.00 eV) and TcTa (2.80 eV) are both higher than that of FIr6, so reverse energy transfer can be well prevented [38,39]. Considering the relatively higher hole injection barrier (1 eV) between PEDOT: PSS (−5.20 eV) and mCPPO1 (−6.20 eV), direct hole injection from PEDOT: PSS to mCPPO1 is very difficult. In this case, TcTa (with moderate energy level of −5.70 eV) would function as hole transfer ladders, which could enhance hole injection, thus broadening RZ and improving carriers’ balance.

To confirm the efficacy of the above design, several single-host or co-host devices with the structure of ITO/PEDOT: PSS/FIr6 (20 wt%): HOST/TmPyPB (60 nm)/LiF (1 nm)/Al (100 nm) were fabricated and compared, where the HOST consisted of one or both of mCPPO1 and TcTa. EL efficiency-current density (*η-J*) characteristics of these devices were depicted in Figure 2, while turn-on voltage, brightness, and efficiency of these devices were summarized in Table 1. Compared with the device based on single-host mCPPO1, the co-host device showed lower turn-on voltage which confirms that the existence of TcTa molecules facilitated the injection of holes from HIL into EML due to the reduced energy barrier. In addition, co-host device showed higher maximum CE and slower efficiency roll-off compared with single mCPPO1 or single TcTa device. The higher CE suggests that bipolar mCPPO1 molecules are beneficial in facilitating the balance of electrons and holes within EML. The reduction rate of efficiency decreased from 40.26% and 50.13% in single-host devices to 30.7% in co-host devices within the current density range of 30–60 mA/cm^2^. Hence, the co-host system consisting of hole-transporting and bipolar material is helpful in improving device performances due to enhanced carriers’ balance and broadening RZ. Besides, several devices with co-host system at various proportions were fabricated and compared, and the optimal proportion of TcTa: mCPPO1 = 1: 1 in the EML was determined to be the optimal proportion (Table 1). As depicted in the inset of Figure 2, no TcTa or mCPPO1 emission was observed, which demonstrated the efficient energy transfer from host to guest.

According to previous investigations, with increasing guest concentration, device efficiency generally increases firstly and then decreases gradually because the decreased intermolecular distance accelerates the unexpected TTA [40]. To determine the optimal doping concentration, a series of devices with the structure of ITO/PEDOT: PSS/FIr6 (x wt%): mCPPO1: TcTa/TmPyPB (60 nm)/LiF (1 nm)/Al (100 nm) were fabricated and examined. Among these devices, as shown in Figure 3a and Table 2, the 20 wt% device displayed the highest performances, with maximum CE, PE, EQE, and brightness of 12.40 cd/A, 7.95 lm/W, 7.54%, and 2993 cd/m^2^, respectively. As depicted in Figure 3b, only FIr6-characteristic emission was observed, demonstrating triplet excitons can be well confined on emitter molecules and that the energy transfer is efficient.

To further enhance carriers’ balance and suppress triplet excitons quenching deep-blue material FCNIrPic was introduced into EML as the second dopant. On the one hand, the relatively higher HOMO energy level of FCNIrPic helps to accelerate holes trapping and the transfer of holes onto FIr6 molecules, thus improving carriers’ balance. On the other hand, due to the energetic proximity of FIr6 and FCNIrPic (2.74 eV) triplet states, energy transfer between these molecules can occur easily, which is helpful in suppressing efficiency roll-off [41]. Besides, the deep-blue emission of FCNIrPic helps to ensure the color purity of the device. Therefore, based on reducing exciton quenching, the total doping concentration of phosphorescent materials was fixed to 20 wt%, a series of co-dopant devices with the EML of FIr6 (20-y wt%): FCNIrPic (y wt%): mCPPO1: TcTa (1: 1) were fabricated and characterized by adjusting y to be 1.0, 2.5, 5.0, 7.5 and 10.0, respectively. With increasing FCNIrPic concentration, the relative intensity of long wavelength peak (487 nm) increases slightly, and the CIE coordinates change slightly (Figure 4b and Table 3), which demonstrated the excellent deep-blue emission characteristic of these devices. EL efficiency increases firstly and then decreases gradually with increasing FCNIrPic concentration, and the optimized efficiency was obtained at the concentration of 5.0 wt%. The maximum CE was increased from 12.40 to 16.74 cd/A, and the maximum brightness of 4121 cd/m^2^ was realized. When FCNIrPic concentration is lower than 5.0 wt%, electrons are more than holes on phosphorescent dopants molecules, causing electron exciton quenching within EML. The efficiency decreases obviously once FCNIrPic concentration is higher than 5 wt%, which can be interpreted as indicating that the triplet energy of FIr6 is more matched with those of mixed hosts (Figure 4d) [42]. However, the highest reduction rate of efficiency decreased from 30.70% in a single-dopant device to 27.89% in co-dopant device within the current density range of 30–60 mA/cm^2^ (Table 3). Therefore, co-dopant devices showed higher maximum CE and slightly slower efficiency roll-off compared with single-dopant device, thus accelerating the injection of holes is more helpful than enhancing energy transfer of excitons in improving device performances.

In addition, as shown in Figure 4a, current density decreases gradually with increasing FCNIrPic concentration, which can be interpreted as the stronger hole-trapping ability of FCNIrPic molecules. Theoretically speaking, the higher energy level increases hole injection, which makes it easier to achieve carriers’ balance and thus improved device performances. To confirm this assumption, we manufactured three hole-only devices with structures: ITO/PEDOT: PSS/mCPPO1: TcTa (1: 1)/TAPC (20 nm)/Al (100 nm), ITO/PEDOT: PSS/FIr6 (20 wt%): mCPPO1: TcTa (1: 1)/TAPC (20 nm)/Al (100 nm), and ITO/PEDOT: PSS/FIr6 (15 wt%): FCNIrPic (5 wt%): mCPPO1: TcTa (1: 1)/TAPC (20 nm)/Al (100 nm), respectively. As shown in Figure 4c, current density decreases with the addition of second dopant FCNIrPic, confirming that FCNIrPic molecules play important role in trapping holes. So, the co-dopant device showed stronger hole trapping ability than single-dopant device. Therefore, device performance was improved by introducing FCNIrPic as the second dopant, because its higher HOMO level accelerates hole injection, and its high triplet energy facilitates energy transfer.

To further enhance device performances, devices A-E were fabricated and compared by adjusting the thickness of ETL from 60 to 100 nm with the step of 10 nm. Current density decreases with increasing ETL thickness (Figure 5a), which indicates that thicker ETL slows down the migration of electrons and delays their injection into EML. As shown in Figure 5b, normalized EL spectra of these devices are strongly thickness-dependent. With increasing ETL thickness, 32 nm redshift (from 458 to 490 nm) of emission peak was observed in devices A and E. According to previous reports, this phenomenon can be interpreted as a microcavity effect caused by the optical path difference of photons [43,44,45].

To confirm this assumption, devices F-K with the thicknesses of ETL at 110, 120, 130, 140, 150 and 160 nm, respectively, were fabricated and measured. With increasing thickness of ETL from 60 to 140 nm, as shown in Figure 6a, the emission spectrum gradually widens while the emission peak shifts gradually from 458 to 528 nm. The maximum redshift of emission peak is about 70 nm. When the thickness of ETL is 150 nm, EL spectrum splits into two peaks located at about 458 and 528 nm, respectively. Further increasing the thickness of ETL to 160 nm caused the collapse of dual peaks into single peak, which recovers to its original position at 458 nm. In addition, as shown in Figure 6b, the shift of emission peak causes the marked change of CIE coordinates with increasing ETL thickness. According to previous investigations, these experimental results could be explained by Fabry-Perot equation [46].
$$\begin{array}{l} \frac{2L}{\lambda }-\frac{\Phi }{2\pi }=m\;\;\;\;\;(m\;\mathrm{is}\;\mathrm{an}\;\mathrm{integer})\\ L=\Sigma (ni*di) \end{array}$$
where *L* is the total optical thickness between anode and cathode, *n* and *d* are refractive index and thickness of each layer, respectively, while *φ* is the sum of the reflection phase difference between cathode and anode. When *m* is an integer, the resonance wavelength *λ* of the resonant cavity can be determined. With the increase of *L*, *λ* will increase gradually and cause the redshift of th EL spectrum. But if *L* is large enough to make *m* enter the next integer, *λ* will blue shift quickly [47]. The above experimental results were strongly consistent with the description of microcavity effect, which confirms the correctness of our inference.

According to SK’s computer model of wide-angle interference and Shogo’s simple model with Fabry-Perot cavity structure, the intensity of EL varies quasi-sinusoidal [48,49]. In addition, *d* values corresponding to the first peak and trough of the EL intensity are a quarter and a half of the emission wavelength divided by the refractive index of the organic material, respectively [50,51,52]. However, when the ETL thickness was increased to 120 nm (Figure 6c), a decreased efficiency was observed. On the one hand, it approaches the trough of EL intensity, and on the other hand, thicker ETL will significantly slow down electron transport and destroy the balance of electrons and holes within EML, thus reducing the recombination probability. Considering both efficiency and spectrum (Figure 5a and Figure 6c), it can be easily identified that the optimal ETL thickness is about 80 nm. Among devices A-E, the optimal device was device C, which realized the maximum CE, PE, and EQE of 20.47 cd/A, 11.96 lm/W, and 11.62%, respectively (Table 4). Therefore, device performances can be improved by adjusting the thickness of ETL to balance carriers’ distribution. Additionally, it can be concluded that EL spectrum is strongly influenced by the thickness of ETL, which can also affect the light outcoupling efficiency by adjusting the resonance wavelength of microcavity. In other words, current efficiency depends on not only the optical factors but also the transport and distribution of carriers. Therefore, the design of device should combine optics and carriers well in order to obtain high performances.

## 3. Materials and Methods

In this study, all organic materials used in the devices were obtained from Luminescence Technology Corp. and used directly. Patterned indium-tin-oxide (ITO)-coated glass (10 Ω/sq) was commercially obtained and used as the anode substrate. The ITO substrates were decontaminated with detergent, rinsed in ultra-purified water, dried at 120 °C for 1 h in oven, and then treated with ambient UV-ozone for 20 min. Poly(3,4-ethylenedioxythiophene): poly(styrene sulfonate) (PEDOT: PSS) was spin-coated onto ITO substrates at the rate of 3000 rpm for 60 s and annealed at 120 °C for 20 min, after which they were transferred into inert-atmosphere glove box. Subsequently, the EML was spin-coated from a chlorobenzene solution onto the PEDOT: PSS layer at the rate of 3000 rpm for 30 s and annealed at 70 °C for 30 min. Then, they were transferred into vacuum chamber to deposit ETL (≤ 5.0 × 10^−6^ Pa) at the rate of 0.1 nm/s. Finally, LiF and Al were deposited in another vacuum chamber (≤ 8.0 × 10^−5^ Pa) at rates of 0.01 and 1.0 nm/s, respectively. The current density-brightness-voltage (*J-B-V*) characteristics and EL spectra were measured by using Brightness Light Distribution Characteristics Measurement System C9920-11. The absorption spectra were measured with CARY 50UV-Vis-NIR Varian spectrophotometer (CA), while the photoluminescence (PL) spectra were recorded on Hitachi F-7000 fluorescence spectrophotometer.

## 4. Conclusions

In summary, we have developed highly efficient solution-processed blue PhOLEDs by constructing co-dopant and co-host structures. Experimental results demonstrated that TcTa molecules function as hole transport ladders and the introduction of FCNIrPic molecules accelerates the transfer of holes onto emitter molecules, thus facilitating carriers’ balance and enhancing light outcoupling efficiency. Furthermore, thicker ETL helps to slow down the migration of electrons, and improved carriers’ balance has been realized by optimizing the thickness of ETL. With increasing thickness of ETL, EL spectrum red shifts firstly and then blue shifts quickly when the thickness of ETL reached 140 nm, which can be explained as microcavity effect between anode and cathode. Finally, the optimal blue phosphorescent device exhibited the maximum CE of 20.47 cd/A, PE of 11.96 lm/W, and EQE of 11.62%. These experimental results validated the efficacy of this device design strategy in preparing high-performance blue PhOLED.

## Figures and Tables

**Figure 1 molecules-27-06882-f001:**
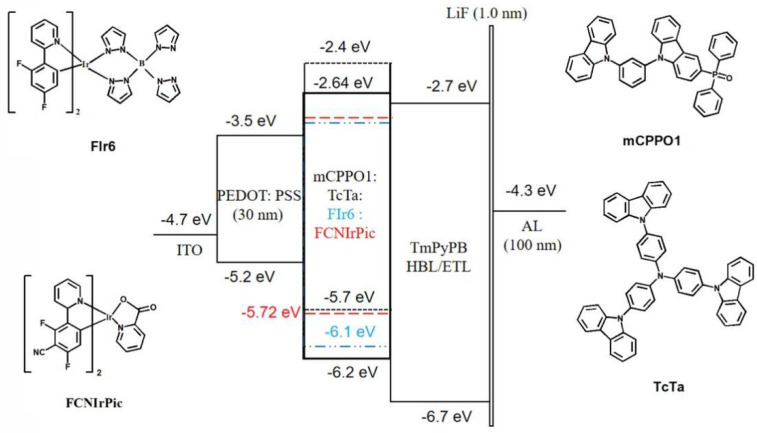
Energy levels diagram of the blue PhOLEDs and molecular structures used in this work, FIr6 (blue dash dot line), FCNIrPic (red dashed line).

**Figure 2 molecules-27-06882-f002:**
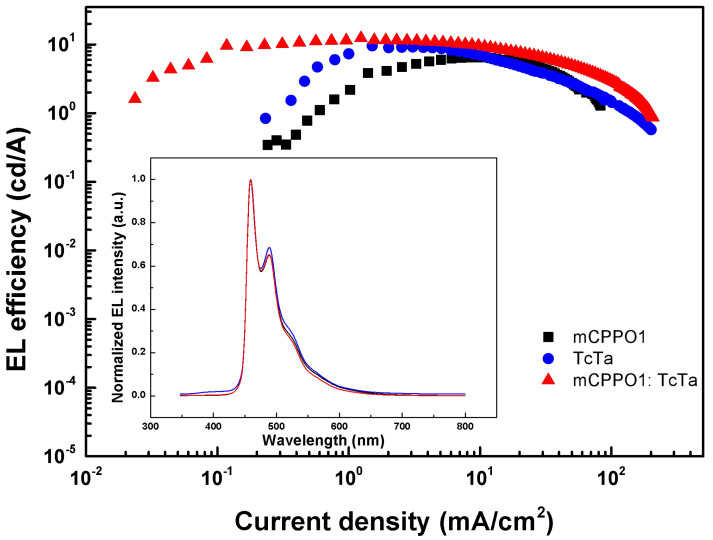
EL efficiency-current density (*η-J*) characteristics of devices with FIr6 doped into different hosts. Inset: Normalized EL spectra of devices with FIr6 doped into different hosts operating at 10 mA/cm^2^.

**Figure 3 molecules-27-06882-f003:**
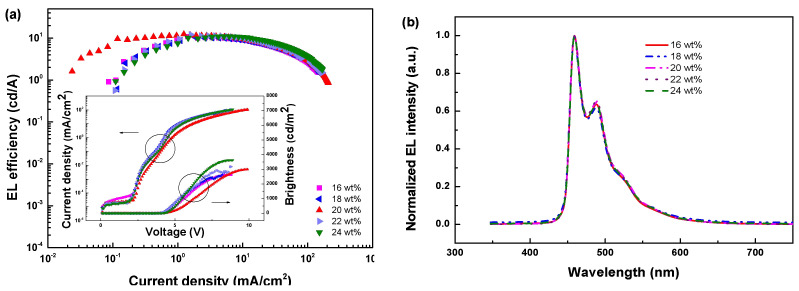
(**a**) EL efficiency-current density (*η-J*) characteristics of co-host devices with FIr6 at different doping concentrations. Inset: Current density-brightness-voltage (*J-B-V*) characteristics of co-host devices with FIr6 at different doping concentrations. (**b**) Normalized EL spectra of co-host devices with FIr6 at different doping concentrations operating at the current density of 10 mA/cm^2^.

**Figure 4 molecules-27-06882-f004:**
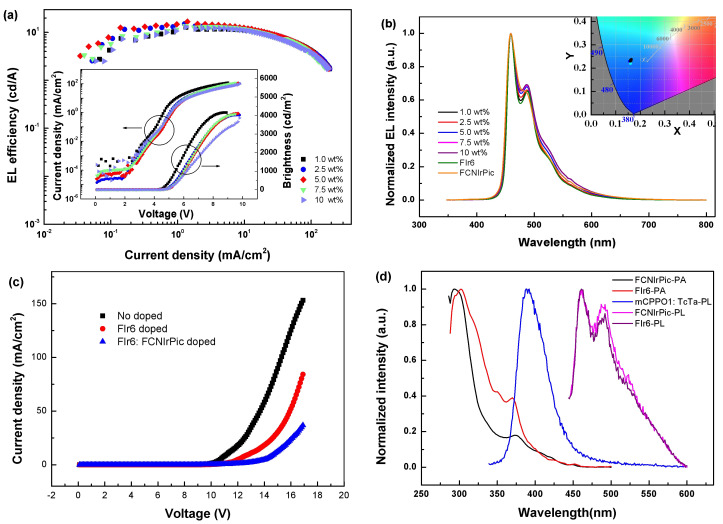
(**a**) EL efficiency-current density (*η-J*) characteristics of co-dopant devices with FCNIrPic at different concentrations. Inset: Current density-brightness-voltage (*J**-B-V*) characteristics of co-dopant devices with FCNIrPic at different concentrations. (**b**) Normalized EL spectra of devices with FCNIrPic at different concentrations operating at the current density of 10 mA/cm^2^. Inset: CIE coordinates of these devices operating at 10 mA/cm^2^. (**c**) Current density-voltage (*J-V*) characteristics of the hole-only devices. (**d**) UV-vis absorption spectra of FIr6 and FCNIrPic and the PL spectra of mixed hosts, FIr6 and FCNIrPic in chlorobenzene at room temperature.

**Figure 5 molecules-27-06882-f005:**
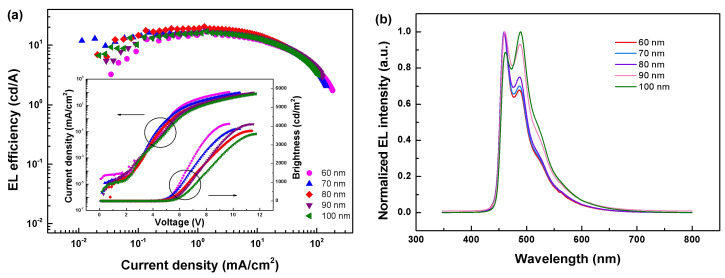
(**a**) EL efficiency-current density (*η-J*) characteristics of co-dopant devices with ETL at different thicknesses. Inset: Current density-brightness-voltage (*J-B-V*) characteristics of co-dopant devices with ETL at different thicknesses. (**b**) Normalized EL spectra of devices with ETL at different thicknesses operating at the current density of 10 mA/cm^2^.

**Figure 6 molecules-27-06882-f006:**
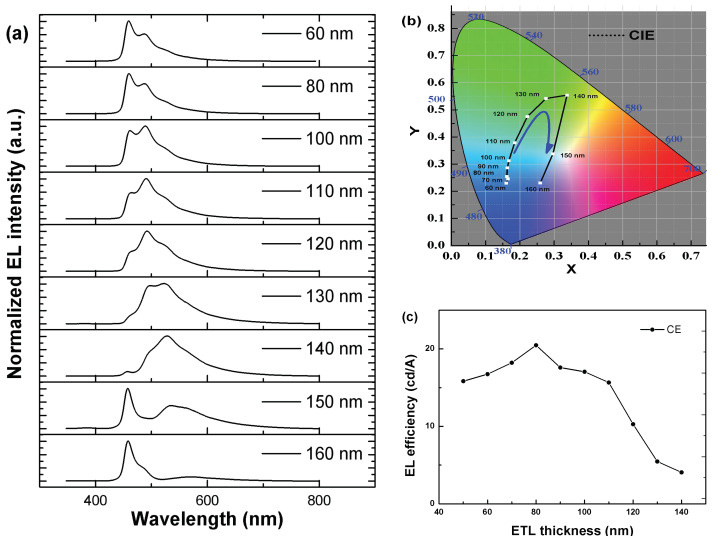
(**a**) EL spectra of OLEDs with various ETL thicknesses. (**b**) CIE coordinates of OLEDs with various ETL thicknesses. (**c**) EL efficiency-ETL thickness characteristics of OLEDs with various ETL thicknesses.

**Table 1 molecules-27-06882-t001:** Key properties of OLEDs with FIr6 doped into different host systems.

Devices	V_turn-on_ (V)	η_c_^a^ (cd/A)	η_p_^b^ (lm/W)	EQE^c^	B^d^ (cd/A)	30–60^e^ (mA/cm^2^)
TcTa	3.5	09.19	7.32	5.73%	1505	3.823/2.284/40.26%
9:1	3.9	10.28	6.73	6.46%	1823	4.603/2.914/36.69%
7:3	3.9	11.08	7.11	7.12%	2256	5.345/3.582/32.98%
5:5	3.7	12.40	7.95	7.54%	2993	6.507/4.509/30.70%
3:7	3.9	10.58	5.63	6.42%	2965	6.648/4.585/31.03%
1:9	4.5	08.61	4.20	5.16%	1478	4.572/2.344/48.73%
mCPPO1	4.5	06.48	3.41	3.89%	1410	4.450/2.219/50.13%

η_c_^a^: maximum current efficiency; η_p_^b^: maximum power efficiency; EQE^c^: maximum external quantum efficiency; B^d^: The data for maximum brightness; 30–60^e^: Current efficiency at 30 and 60 mA/cm^2^ and efficiency roll-off rate within the current density range of 30–60 mA/cm^2^.

**Table 2 molecules-27-06882-t002:** Key properties of co-host devices with FIr6 at different doping concentrations.

FIr6 wt%	V_turn-on_ (V)	η_c_^a^ (cd/A)	η_p_^b^ (lm/W)	B^c^ (cd/A)	EQE^d^
16	3.9	10.89	7.44	2615	6.71%
18	3.8	11.65	8.32	2891	7.12%
20	3.7	12.40	7.95	2993	7.54%
22	3.8	11.37	8.66	3177	7.43%
24	3.9	10.96	6.84	3626	6.79%

η_c_^a^: maximum current efficiency; η_p_^b^: maximum power efficiency; B^c^: The data for maximum brightness; EQE^d^: maximum external quantum efficiency.

**Table 3 molecules-27-06882-t003:** Key properties of co-dopant devices with FCNIrPic at different concentrations.

Devices	V_turn-on_ (V)	η_c_^a^ (cd/A)	η_p_^b^ (lm/W)	EQE^c^	B^d^ (cd/A)	CIE_x,y_^e^	30–60^e^ (mA/cm^2^)
FIr6 (20 wt%)	3.7	12.40	7.95	07.54%	2993	(0.155,0.229)	6.507/4.509/30.70%
1 wt%	4.0	12.74	8.52	07.63%	4167	(0.160,0.220)	8.380/6.043/27.89%
2.5 wt%	4.3	15.08	9.29	09.17%	4201	(0.157,0.232)	8.699/6.129/29.54%
5.0 wt%	4.3	16.74	9.93	10.05%	4121	(0.159,0.234)	8.639/6.015/30.37%
7.5 wt%	4.3	13.63	8.08	07.94%	3728	(0.160,0.230)	8.261/5.764/30.23%
10 wt%	4.3	12.98	7.84	07.68%	3930	(0.164,0.238)	7.898/5.408/31.53%

η_c_^a^: maximum current efficiency; η_p_^b^: maximum power efficiency; EQE^c^: maximum external quantum efficiency; B^d^: The data for maximum brightness; 30–60^e^: Current efficiency at 30 and 60 mA/cm^2^ and efficiency roll-off rate within the current density range of 30–60 mA/cm^2^.

**Table 4 molecules-27-06882-t004:** Key properties of OLEDs with ETL at different thicknesses.

Devices	V_turn-on_(V)	η_c_^a^ (cd/A)	η_p_^b^ (lm/W)	EQE^c^	B^d^ (cd/A)	CIE_x,y_^e^
50	3.6	15.83	9.57	9.08	2974	(0.160, 0.230)
60	4.3	16.74	9.93	10.05	4121	(0.165, 0.245)
70	3.7	18.21	10.91	10.41	3853	(0.162, 0.254)
80	4.0	20.47	11.96	11.62	3751	(0.163, 0.286)
90	4.4	17.59	9.53	9.38	4110	(0.168, 0.312)
100	4.4	17.05	8.78	8.61	3580	(0.185, 0.379)
110	4.0	15.66	8.95	6.97	3257	(0.222, 0.475)
120	4.3	10.28	5.98	3.89	2206	(0.276, 0.542)
130	4.3	5.46	3.12	1.82	1018	(0.338, 0.554)
140	4.5	4.07	2.32	1.27	679	(0.296, 0.338)
150	4.8	1.74	0.99	0.86	263	(0.259, 0.230)
160	4.9	1.15	0.62	0.77	188	(0.165, 0.245)

η_c_^a^: maximum current efficiency; η_p_^b^: maximum power efficiency; EQE^c^: maximum external quantum efficiency; B^d^: the data for maximum brightness; CIE_x,y_^e^ Commission Internationale de I’Eclairage coordinates (CIE_x, y_) at 10 mA/cm^2^.

## Data Availability

The data presented in this study are available in the article.

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
