# Peer review of "Highly Efficient Solution-Processed Blue Phosphorescent Organic Light-Emitting Diodes Based on Co-Dopant and Co-Host System"

_molecules, 2022, doi:10.3390/molecules27206882_

Round 1
Reviewer 1 Report
OLEDs are very attractive for displays and lighting. However, the efficiencies and the fabrication cost are required to be optimized for mass production. In this contribution, the authors optimized the device structure of the solution-processed phosphorescent OLEDs and found the dual-emitter strategy and the thickness of the ETL both played roles in ruling the EQEs and roll-offs. To make it clearer to the readers and more acceptable for publication, the following concerns should be addressed.
1. The fabrication procedures, especially for the solution-processed EMLs, should be clearly clarified.
2. As shown in Line 121, "Reduction rate of efficiency ..." is not a common expression. It seems related to the efficiency roll-off. Please make it clear to the readers. The similar issue appeared in Line 136, i.e., "Efficiency roll-off rate".
3. The y-axis of Figure 4d was not correctly indicated. Moreover, Figure 6a was not located and displayed properly.
4. To understand the energy transfer in the EML, it would be better to add the PL spectra of FIr6 and FCNIrpic into Figure 4d.
Overall, some minor revision is required before the acceptance could be recommended.
Author Response
Reviewer #1:
- The fabrication procedures, especially for the solution-processed EMLs, should be clearly clarified.
Our Reply: We thank the reviewer very much for the invaluable comment. According to the reviewer’s suggestion, we have added the fabrication procedures of the solution-processed EMLs. “Subsequently, the EML was spin-coated from a chlorobenzene solution onto the PEDOT: PSS layer at the rate of 3000 rpm for 30 seconds and annealed at 70 ℃ for 30 minutes.” The detailed revisions can be found in ‘Marked Revised Manuscript’.
- As shown in Line 121, "Reduction rate of efficiency ..." is not a common expression. It seems related to the efficiency roll-off. Please make it clear to the readers. The similar issue appeared in Line 136, i.e., "Efficiency roll-off rate".
Our Reply: We thank the reviewer very much for the valuable advice. In order to better understand that reduction rate of efficiency is (ηc30-ηc60)/ηc30, we have added two sets of data of current efficiency at 30 and 60 mA/cm2 in Table 1 and Table 3. The detailed revisions can be found in ‘Marked Revised Manuscript’.
- The y-axis of Figure 4d was not correctly indicated. Moreover, Figure 6a was not located and displayed properly.
Our Reply: We thank the reviewer very much for giving us valuable suggestions. The y-axis of Figure 4d ‘Normalized EL Intensity (a.u.)’ has been revised to be ‘Normalized Intensity (a.u.)’. And Figure 6a has been located and displayed properly. The detailed revisions can be found in ‘Marked Revised Manuscript’.
- To understand the energy transfer in the EML, it would be better to add the PL spectra of FIr6 and FCNIrpic into Figure 4d.
Our Reply: We thank the reviewer very much for giving us valuable suggestions. According to the reviewer’s suggestion, we have added the PL spectra of FIr6 and FCNIrpic to Figure 4d. The detailed revisions can be found in ‘Marked Revised Manuscript’.

Reviewer 2 Report
This manuscript reported highly efficient solution-processed blue PhOLEDs with co-dopant and co-host structures. Adding TcTa molecules and FCNIrPIC molecules can facilitate carriers’ balance and enhance light outcoupling efficiency. The properties of OLEDs were well-studied. The optimal blue phosphorescent device exhibited the maximum CE of 20.47 cd/A, PE of 11.96 LM/W, and EQE of 11.62%. A large amount of data is included in this manuscript, therefore, I would like to recommend its publication after minor revisions. Comments of this paper are shown below:
1. When the ETL thicknesses were 90 nm, 100 nm and 110 nm, the efficiency decreased slowly, however, when the ETL thickness was 120 nm, the dramatic decreased efficiency was observed. If possible, the authors might give more explanation.
2. The maximum EQE was obtained when the ETL thickness was 80 nm. Hence, it may be better to give the EL spectrum of OLEDs with ETL thickness of 80 nm in Figure 6(a).
3. References should be unified format.
Author Response
Reviewer #2:
- When the ETL thicknesses were 90 nm, 100 nm and 110 nm, the efficiency decreased slowly, however, when the ETL thickness was 120 nm, the dramatic decreased efficiency was observed. If possible, the authors might give more explanation.
Our Reply: We thank the reviewer very much for the invaluable and thoughtful comment. When the ETL thickness was 120 nm, dramatic decrease in efficiency was observed. Our explanation is “On the one hand, the thickness of ETL approaches the trough of EL intensity, and on the other hand, thicker ETL will significantly slow down electron transport and destroy the balance of electrons and holes within EML, thus reducing the recombination probability.” The detailed revisions can be found in ‘Marked Revised Manuscript’.
- The maximum EQE was obtained when the ETL thickness was 80 nm. Hence, it may be better to give the EL spectrum of OLEDs with ETL thickness of 80 nm in Figure 6(a).
Our Reply: We thank the reviewer very much for the valuable advice. According to the reviewer’s suggestion, we have replaced the EL spectrum of OLEDs with ETL thickness of 90 nm with that of 80 nm in Figure 6(a). The detailed revisions can be found in ‘Marked Revised Manuscript’.
- References should be unified format.
Our Reply: We thank the reviewer very much for the instructive suggestion. Reference uses the same Palatonp Linotype font as the text and unified format. The detailed revisions can be found in ‘Marked Revised Manuscript’.

Reviewer 3 Report
Herein, the authors have used double-host systems in phosphorescent OLEDs to attempt to maximize quantum efficiency. They did observe an improvement in device performance by adding TaTc as additional hole transport material, and optimized the parameters for device fabrication.
The work is publishable in Molecules, after a thorough check of the English to make sure that sentence structure and syntax is consistent. Also Figure 6a in the manuscript overlaps with Table 4, it will need to be placed properly in the final version.
Author Response
Reviewer #3:
The work is publishable in Molecules, after a thorough check of the English to make sure that sentence structure and syntax is consistent. Also Figure 6a in the manuscript overlaps with Table 4, it will need to be placed properly in the final version.
Our Reply: We thank the reviewer very much for the invaluable comment. We have checked the English to make sure that sentence structure and syntax are correct, and properly placed the position of Figure 6a and Table 4 in the manuscript. The detailed revisions can be found in ‘Marked Revised Manuscript’.

Reviewer 4 Report
In the submitted paper the authors present co-host system composed of bipolar material 9-(3-(9H-carbazol-9-yl)phenyl)-3-(diphenylphosphor-yl)-9H-carbazole (mCPPO1) and hole-transporting material 4,4',4"-Tris(carbazol-9-yl)triphenylamine (TcTa) and introducing the second dopant Bis(3,5-difluoro-4-cyano-2-(2-62 pyridyl)phenyl-(2-carboxypyridyl)iridium(III) (FCNIrPic) to facilitate carriers’ balance as well as reduce exciton quenching, thus improving the efficiency of PhOLED. They observed an interesting phenomenon caused by the microcavity effect between anode and cathode. Furthermore, with increasing thickness of ETL, the peak position of the electroluminescence (EL) spectrum red-shifts gradually.
My remarks & comments are the following:
1. In the Introduction, the abbreviations used should be explained, for chemical compounds used as examples (from line 49 to line 55), at least their chemical name should be given to be able to refer to their structure.
2. Line 69: “Experiments” I think it was supposed to be the title of another section (2) of the type: Materials and Methods, which is just missing on line 70. This line also does not specify which companies the organic materials were purchased from.
3. Line 110: Individual layer thicknesses are missing.
4. Fig. 3a is hardly legible, while Fig. 6a has moved over to table 4.
Author Response
Reviewer #4:
- In the Introduction, the abbreviations used should be explained, for chemical compounds used as examples (from line 49 to line 55), at least their chemical name should be given to be able to refer to their structure.
Our Reply: We thank the reviewer very much for the invaluable and thoughtful comment. The abbreviations used in the chemical compounds in the introduction have been explained. The chemical name of m-MTDATA is “4,4’,4’’-Tris(N-3-methylphenyl-N-phenyl-amino)-triphenylamine”; The chemical name of Ir(ppy)3 is “fac-Tris(2-phenylpyridine)iridium(III)”; The chemical name of DCM2 is “4-(Dicyanomethylene)-2-methyl-6-julolidyl-9-enyl-4H-pyran”. The detailed revisions can be found in ‘Marked Revised Manuscript’.
- Line 69: “Experiments” I think it was supposed to be the title of another section (2) of the type: Materials and Methods, which is just missing on line 70. This line also does not specify which companies the organic materials were purchased from.
Our Reply: We thank the reviewer very much for giving us valuable suggestions. We have moved "Experiments" to the third part of Materials and Methods. Moreover, we specified that these materials were obtained from Luminescence Technology Corp. The detailed revisions can be found in ‘Marked Revised Manuscript’.
- Line 110: Individual layer thicknesses are missing.
Our Reply: We thank the reviewer very much for the instructive suggestion. According to the reviewer’s suggestion, we added the thickness of PEDOT: PSS layer in Figure 1. The detailed revisions can be found in ‘Marked Revised Manuscript’.
- Fig. 3a is hardly legible, while Fig. 6a has moved over to table 4.
Our Reply: We thank the reviewer very much for the invaluable comment. For the illegible problem in Figure 3a, we redrew it to make it clearer. We properly placed the position of Figure 6a and Table 4 in the manuscript. The detailed revisions can be found in ‘Marked Revised Manuscript’.
